# Can we normalise developmentally appropriate health care for young people in UK hospital settings? An ethnographic study

Tim Rapley,[1,2] Albert Farre,[3] Jeremy R Parr,[4] Victoria J Wood,[5] Debbie Reape,[6] Gail Dovey-Pearce,[6] Janet McDonagh,[7,8] The Transition Collaborative Group

For numbered affiliations see end of article.

**Correspondence to**
Tim Rapley;
tim.rapley@northumbria.ac.uk

## ABSTRACT

**Objective** The WHO has argued that adolescent-responsive health systems are required. Developmentally appropriate healthcare (DAH) for young people is one approach that could underpin this move. The aim of this study was to explore the potential for DAH to become normalised, to become a routine, taken-for-granted, element of clinical practice.

**Design** Qualitative ethnographic study. Analyses were based on procedures from first-generation grounded theory and theoretically informed by normalisation process theory.

**Setting** Two tertiary and one secondary care hospital in England.

**Participants** 192 participants, health professionals (n=121) and managers (n=71) were recruited between June 2013 and January 2015. Approximately 1600 hours of non-participant observations in clinics, wards and meeting rooms were conducted, alongside 65 formal qualitative interviews.

**Results** We observed diverse values and commitments towards the care of young people and provision of DAH, including a distributed network of young person-orientated practitioners. Informal networks of trust existed, where specific people, teams or wards were understood to have the right skill-mix, or mindset, or access to resources, to work effectively with young people. As young people move through an organisation, the preference is to direct them to other young person-orientated practitioners, so inequities in skills and experience can be self-sustaining. At two sites, initiatives around adolescent and young adult training remained mostly within these informal networks of trust. At another, through support by wider management, we observed a programme that sought to make the young people's healthcare visible across the organisation, and to get people to reappraise values and commitment.

**Conclusion** To move towards normalisation of DAH within an organisation, we cannot solely rely on informal networks and cultures of young person-orientated training, practice and mutual referral and support. Organisation-wide strategies and training are needed, to enable better integration and consistency of health services for all young people.

## INTRODUCTION

The health of young people, defined by the WHO as any person between ages 10 and 24

### Strengths and limitations of this study

► This is the first ethnographic study addressing the provision of developmentally appropriate healthcare for young people.
► Exploring the provision of healthcare for young people over time, within specific teams and specialities, as well as across an organisation, enabled us to explore the diversity of ways that healthcare for young people is currently delivered within and across professionals, teams, wards and organisations.
► We only focused on three different organisations and all these organisations had a reputation for undertaking research on the care young people, so may represent examples of 'good practice'.

years,[1] is a neglected yet pressing global issue[2] affecting the largest generation in history.[3] Youth-friendly healthcare,[4] promoted as a means to improve health services for young people, has underpinned quality of care and policy frameworks.[5–7] However, as highlighted by the WHO, there is the need to move from the ad hoc provision of youth-friendly healthcare services—often embedded in specific locations or teams—to adolescent-responsive healthcare systems.[8] All the aspects of health and social care that young people engage with, the range of providers, organisations and policies, need to more responsive to and aligned with the care of young people. Other work has also noted that health system-level strategies are needed to further develop and improve healthcare for young people.[9 10] Healthcare providers need to respond to young people's changing developmental needs in a manner that is consistent, universal and provided across healthcare settings. Such adolescent-responsive health systems should be flexible, and should focus on a broad range of aspects of healthcare provision, from the frontline

**Box 1.  The five dimensions of developmentally appropriate healthcare (DAH)**

Dimensions of DAH for young people[9–11]

**Biopsychosocial development and holistic care:** a focus on biopsychosocial development rather than chronological age, with routine biopsychosocial developmental assessment and approach to the young person adjusted accordingly.

**Acknowledgement of young people as a distinct group:** the recognition that their specific needs—in terms of informational resources, services, spaces, pathways and required competencies of staff—are distinctly different to those of younger children and older adults as a result of their developmental status.

**Adjustment of care as the young person develops:** the need for flexibility to acknowledge the biopsychosocial developmental changes over time and the potential for regression in relapsing health conditions.

**Empowerment of the young person by embedding health education and health promotion**: that knowledge and skills training for young people is embedded into routine clinical practice to enable them to gradually become more autonomous with respect to the care of their own health as they grow up. Services need to be designed so as to nurture and support such skill development.

**Interdisciplinary and interorganisational work:** a focus on continuity of care, coordination, consistency and communication across agencies. Connecting health, education, employment, social, voluntary agencies at a clinical and system level.

(such as how healthcare professionals communicate with young people) through to the higher levels of healthcare provision (such as how services are planned and commissioned).

Developmentally appropriate healthcare (DAH) for young people[11–13] is one concept that could underpin an adolescent-responsive healthcare system (see Box 1).

In contrast to the more service-focussed nature of the youth-friendly health service,[14] DAH addresses the clinical approach to individual young people and specifically recognises the changing developmental needs of young people and the role of healthcare in addressing and supporting young people. DAH focuses on biopsychosocial development rather than chronological age. Chronological age is recognised to be a poor indicator of developmental status particularly in the context of a long-term health condition.[15] Young people make this journey to adulthood in their own way; young people's development does not have a fixed period attached to it. Many developmental milestones are met after reaching the legal age of adulthood.[16 17]

Services underpinned by DAH have been reported nationally and internationally as a potential key mechanism to improve health outcomes for young people.[3 18] Increasing knowledge about the development of young people[19 20] offers unprecedented opportunities for service improvement. In the context of suboptimal provision of healthcare for young people,[3 8] DAH offers the potential to transform traditional models of healthcare delivery into adolescent-responsive healthcare systems. However, the concept of DAH has been operationalised in a range of ways in the medical literature[11] and is

understood in different ways by clinicians and managers in the National Health Service (NHS).[12] In this study, theoretically informed by normalisation process theory (NPT),[21] we investigated the potential for normalisation of DAH within three UK hospitals. NPT identifies, characterises and explains aspects of individual and collective behaviour shown to be important in empirical studies of the introduction, embedding and integration of change.[21] Normalisation is achieved when a technique, technology or organisational change becomes a routine and taken-for-granted element of clinical practice.

## METHODS

This ethnographic study was conducted across three hospitals in two regions of England (a district general hospital, a large paediatric tertiary hospital and a large adult tertiary hospital) all in urban settings outside London. All hospitals had a history of championing research and innovative service provision for young people. At the start of the fieldwork one site—the General Hospital—was developing a policy about DAH. This had emerged from a formal, organisationally supported, group that focused on the care of young people, that was initially formed to focus on transition. This strategy group comprised of managers and senior clinicians. At another site—the Paediatric Hospital—there was no explicit policy on DAH, but one focused on transition. They also had a formal, organisationally supported, group focused on young people. However, thinking about DAH was being driven by enthusiasts alone. Finally, the third site had no policy on DAH, transition or young people. The sole organisation-wide initiative is an informal interest group focused on young people issues. Health professionals, recruited through six medical and surgical specialties (Diabetes, Emergency Care, General Paediatrics, Outpatients, Rheumatology and Trauma and Orthopaedics), were chosen to represent the heterogeneous services found in UK NHS hospitals. Managers were recruited at each site when their roles were relevant to the provision of services for young people in paediatrics and/or adult care.

Data collection took place over three phases, between June 2013 and January 2015. Recruitment was initially mediated through gatekeepers. Participants were purposively sampled, initially through maximum variation sampling and then refined through snowball sampling and theoretical.[22] Participants were approached to take part in the study using a variety of methods including face-to-face, telephone and email. A total of 192 participants (professionals and managers) were recruited (table 1).

Approximately 1600 hours of non-participant observations were conducted by two researchers (AF and VJW), alongside 65 formal qualitative interviews. Researchers had specific training and extensive experience in conducting qualitative research and had no relationship with the participants. Observations were conducted within and across a wide variety of hospital spaces—including clinics, wards and meeting rooms—depending

**Table 1** Recruitment for the study by site, type of staff and method of data collection

| | | District general hospital | Paediatric tertiary hospital | Adult tertiary hospital | Total |
|---|---|---|---|---|---|
| Participants observed only | Health professionals | 53 | 22 | 10 | 85 |
| | Managers | 43 | 0 | 12 | 55 |
| | Total | 96 | 22 | 22 | 140 |
| Participants interviewed only | Health professionals | 0 | 14 | 9 | 23 |
| | Managers | 0 | 5 | 2 | 7 |
| | Total | 0 | 19 | 11 | 30 |
| Participants interviewed and observed | Health professionals | 8* | 4 | 1 | 13 |
| | Managers | 5* | 1 | 3 | 9 |
| | Total | 13 | 5 | 4 | 22 |
| Total number of participants | | 109 | 46 | 37 | 192 |

*Participants interviewed twice.

on the nature of the session and/or the professionals involved. They were recorded in contemporaneous fieldnotes. Participants were selected for formal interviews to follow-up specific issues emerging from observation sessions and/or prior interviews. Formal interviews (average length, 45 min) were conducted face-to-face on a one-to-one basis and were audiorecorded, transcribed, edited to ensure respondents anonymity and then analysed alongside anonymised fieldnotes. Initial topic guides were designed for clinicians and managers irrespective of setting and evolved during the course of data collection, allowing for tailoring and gradual integration of a variety of follow-up issues and topics of relevance to specific roles, settings, specialities or areas. Observation and interviews focused on the organisation of services for young people, including (intra-/inter-) organisational, team and individual aspects of provision, training and support, and patient involvement as well as emerging topics identified through concurrent data analysis.

All analysis was conducted according to the standard procedures of rigorous qualitative analysis by AF, VW and TR.[23] We used procedures from first-generation grounded theory—coding, constant comparison, memoing[24]—and from analytic induction, deviant case analysis.[25] Sampling, data collection and analysis occurred concurrently, so that issues raised in earlier phases of fieldwork were explored subsequently to enable conceptual saturation.[26] We undertook independent coding and cross checking, team data sessions and member validation with some of the participants in the fieldwork. The analysis was assisted by QSR NVivo 10 software and theoretically informed by NPT.[21] In presenting the analysis, we have drawn on interview quotes, over excerpts from fieldnotes, as they offer the reader rapid access to the key analytic ideas.

### Patient and public involvement
This study was part of a larger longitudinal programme of research focused on the commissioning and provision of healthcare services for young people.[27 28] The research questions for the programme were initially informed by engagements with a pre-existing young people's advisory group, third-sector voluntary agencies and pupils from a school for young people with physical impairments and students from a living skills course at a college. A young people's advisory group was established as part of programme and supported all studies within the programme. The group advised on practical (eg, recruitment) and conceptual issues (eg, design of study, discussion of key findings and concepts) for this specific study. Young people led on aspects of dissemination, including the production of a video about DAH, for this study.

### RESULTS
We identified diverse values and commitments towards the care of young people and provision of DAH across organisations, specialities and staff. We observed a range of informal cultures of good practice implementing core elements of DAH, alongside formal and informal cultures of training around DAH and the provision of care of young people.

### Diverse values and commitments
When discussing the provision of care for young people, many focused on the need to create a more age appropriate environment, in terms of physical space, the visual and material culture of waiting room and wards. Others, especially those who worked regularly with young people, discussed features such as appropriate communication, confidentiality and a more holistic focus. As one manager noted:

> The young people have told us that, um, we need, they're not bothered about where they are seen so, as in, what the building's look like or what the clinic room looks like. They want to make sure that they see somebody who's interested and who knows what their disease is like, but also has an awareness of all

the other stuff that's going on when they're 16 to 18. (Manager, General Hospital)

At this hospital, involvement was considered to be part of patient experience, and young people were involved in such activities as the, attendance at governance meetings and training of staff. Young people's experiences were an important influence on this manager's understanding of 'appropriate' service provision.

There were conflicting views on the value and worth of enacting DAH across the organisations. The numbers of young people accessing health services were often portrayed as small, so in organisational terms they were 'just below the radar'. In contrast to older, especially elderly patients, they were also seen as 'very rarely unwell'. This led some to question whether professionals should be adjusting their practices or offering distinct, tailored, services, as there are always competing demands for resources, time and expertise.

It's small enough [numbers] that if you don't, if you don't buy into it, there's plenty to be getting on with the other 90%. And everybody's jobs are so frantic that you could easily do a very good job with that 90% who are 25, 26 plus. … So, you could actually ignore these young people completely. (Manager, Adult Hospital)

Questions of legitimacy and buy-in, or rather, enrolment, are central to individual, team and organisational change. In part, this lack of capacity and willingness seems to be compounded by the liminal status of young people within the organisational and professional culture of the UK healthcare system:

Adults don't really want them because they are too young and the paediatricians don't really want them because they are too old. (Health Professional, General Hospital)

With the exception of charity-supported oncology wards, adolescent only wards, or spaces within wards, were rare and were often subject to dissolution if there were competing demands. None of the sites had a senior clinical or management lead with responsibility for young people's service development.

However, across all the sites, we did observe a complex network of young person-orientated practitioners, people acting as young people 'enthusiasts', or 'champions', within specific clinical or management teams in either a formal or informal capacity. Some specific services and practices were organised around the care of young people. However, relying on the enthusiasm and willingness of specific individuals can become problematic.

Several people have left, who have been very senior members and very adolescent minded and have been replaced by either rotational posts or part-time posts, so that continuity within our team and the wealth of expertise has been impacted on significantly. (Health Professional, Paediatric Hospital)

A focus on champions raises key questions about the sustainability of services for young people over time.

### Informal cultures of good practice

Across all three hospitals, we observed local cultures—in teams, clinics, wards and meetings—where professionals attempted to enact a philosophy of care towards young people. They were driven by an awareness of how approaching young people in a different way can mean that young people 'might take the right messages away, might not end up neglecting their health needs, damaging themselves' (Manager, General Hospital). For these professionals, working with young people required a specific mindset and skill-set.

At its simplest level this often involved the ability of the professional, or the multidisciplinary team, to communicate effectively, listening to young people about their health needs and asking them questions, about their broader psychosocial situation. Care is contextualised by gathering information about a young person's life, including educational, vocational, social, friendship and family issues; and exploring risk and resilience factors. Such information is used by teams to generate an appropriate context for effective communication, inform interventions and to organise consultations, including offering appropriate health education and support for self-management.

Providing an appropriate service for young people is often a deeply rooted value for these individuals and groups. The 'You're Welcome Quality Criteria'[6] were generally well known, including core issues such confidentiality and consent, joined-up working, transition and accessibility. Within the context of adult care, failure to attend appointments was a prevalent issue. Ways of dealing with this varied and we were told about what was referred to as, 'a softer approach to the DNAs (did not attend)'.

The organisation ruling of one strike and you're out, we don't adhere to, so we will give them multiple attempts to come into clinic … we don't actually put them as a DNA, because they just booked in (the consultants) calendar but we don't actually book it on the system so they don't officially come as a DNA … So I'll make informal appointments with the young people and then, when they arrive we book them into clinic, so that way they don't DNA. (Health Professional, Adult Hospital)

Working creatively within the existing norms, rules and resources was characteristic of those who believed that young people needed to be recognised as a group with specific needs and approaches.

We observed across all the organisations 'lots of great pockets of work' as some specialties, teams, people or spaces offered very strong young person-orientated care. However, not all people or services felt it relevant to make 'special arrangements', but chose instead to treat young people like 'an ordinary patient'. Alongside this,

the uneven distribution of resources within and across specialties created inequities of care.

> There are … areas in the hospital who, because they have more funding or they're funded in a different way, they might have a youth worker because it's part of their team and just for their team. They might have a psychologist who is just part of their team. … it very much depends on what speciality you're unfortunate to fall into, depending on what illness you've got as to what service you then get. …. (Health Professional, Paediatric Hospital)

However, resources were not the only source of inequities. In part, the inequities in skills and experience across the organisations seemed to be self-sustaining within organisations. Those with an interest, the 'enthusiasts', were embedded in an informal network of care.

> we've now got a group of interested people across the (organisation). So if a young person comes to me and they've got a, a joint problem, but they've also got a bowel problem, I know which bowel consultant and which bowel nurse will be the most appropriate to send them to. … Um, so we've got a good group of people across the (organisation) that we can actually send these youngsters to who've got more awareness of the issues that they could have. (Health Professional, General Hospital)

An informal knowledge economy of young person-orientated practitioners and practices existed within the organisations. These were networks of trust. These referrals helped to create, sustain and reinforce the network over time. This practice also existed across organisations, especially in terms of the transfer of young people to adult services. Young person-orientated practitioners referred to other young person-orientated practitioners—in this way, they worked to actively avoid referring young people to those they felt maybe less young person-orientated. These people then gained less practical experience with managing these patients, and so had less chance to reappraise their values and commitment to working with young people in new ways, as well as to develop the right skill-mix.

An informal network also existed in terms of spaces. At each site, at least one specific ward was known to offer more young person-orientated care. They were seen as repositories of knowledge and skills, able to advise on or manage potentially challenging behaviour.

> We were getting so much enquiries regarding adolescents from the other wards, even just for the basics. … So, they would ring us and say, 'We can't get them out of bed in the morning'. You know, 'they just want to stay in bed all the time and they don't want to interact with anything'. So we would say, 'Well then you have to be stern, you know, you have to tell them, "This is the plan", you have to do a contract with them and agree with them that if they get over this time, then

they can do this at this time'. (Health Professional, Paediatric Hospital)

It is not only that 'just different wards have a different tolerance', but rather that different wards and teams, developed, over time, different understanding and a different sense of what was legitimate work, as well as developmentally appropriate skills, competencies and routines.

> In some situations, we've had patients on our general wards where the parent has wanted to stay. And we, my nurses, would find that very strange. But actually, in oncology, that would not be strange at all. Because … (they) would be used to that, even a patient could be 22, 23 and still may want their mum. But they're not exposed to that in the main wards. (Health Professional, Adult Hospital)

Exposure to working with young people was central to adjusting expectations, enabling them to longer see young people as having 'strange' requests or being particularly 'difficult' to work with. Exposure offered a chance to develop new skills.

## (In)formal cultures of training

Within each of the organisations, there were formal and informal groups at which the young person-orientated enthusiasts met and supported each other collectively. Essentially, these groups were a collective effort to promote initiatives to raise awareness across the organisations, create change, offer support and, importantly, learn from each other outwith their team, area or specialty.

> I have nothing in writing in my job plan that says I specialise in young people. … Nobody said, 'If you want to be a young adult person, you need to go on this (training course)'. It's just something that I became aware of through organisations or talking to people. So, it's all quite ad-hoc rather than really, really planned. And it's just really by hearsay and talking to people and networking throughout (this organisation) over many years. (Health Professional, Paediatric Hospital)

Without any formalised professional routes available, the local, regional and national special interests groups became a central resource for supporting adolescent and young adult health training across the organisations.

At two hospitals, the only initiatives involving training around young people originated from their respective special interest groups, in the form of annual study days. At one of these hospitals, there was a policy initiative explicitly around transition, yet no specific training had been organised. At the other, development of a formal policy was said to be 'not a priority for the organisation'.

> Priorities are the front door, A&E (Accident and Emergency), Clinical Decisions Unit, waiting times and, it's those things that they are being judged on.

Interviewer: Why do you think this is not a priority at all?

Just because they've got bigger fish to fry. … But it's not a priority for the (organisation) because of all the other things by which they are measured. And young people's care isn't on that list. (Manager, Adult Hospital)

In the current context of the factors that drove organisational change at this site, creating further engagement and buy-in from senior management were not seen as practical solution. Issues about the care of young people remained focused in the informal, organisation-wide group, of young people's enthusiasts. As we discovered, not everyone interested in the care of young people in that organisation was aware of the existence of that group. Even those within the group were often unaware of the range of young person-orientated initiatives that where occurring within their organisation.

At the remaining hospital specific training around the topic of DAH was observed being planned and delivered. A DAH strategy emerged as the result of the work of key people who sat on a transition strategy group. The group comprised of managers and health professionals, who met bimonthly. There was strong cross-over between managerial and clinical levels and they worked to actively foster communication and create connexions across services. The ideas emerged from the local special interest group, but the dissemination was targeted well beyond that group. Part of this involved looking where change was currently occurring within the organisation, alongside the broader national agendas on young people's health, in order to harness that momentum and get people involved.

The focus of the strategy and training was on organisational level factors (eg, staff appraisal including training goals around young people; provision of age-banded clinics), clinic and consultation level factors (eg, signpost sexual health, drug and alcohol services; copying clinic letters to young people) and training and awareness factors (eg, adolescent development; confidentiality). This programme of training had senior management support, although initially only from child health, alongside access to resources. Using money to 'back fill' was seen as a key component in the success of the roll-out of training, as without this departments would not only be unwilling but also unable to release staff for training.

Last year we did, um, we did ten days, so ten individual day sessions for training in adolescent, basic adolescent health. Basically to increase awareness across the (organisation) so as to make sure it wasn't just the, the chronic illness patients that were being looked at … but it was the patients coming through A&E, coming through X-ray … just to get them aware of what a young person's needs are and why they're different to being an adult. (Health Professional, General Hospital)

Central norms and practices of good, everyday, care for young people were distributed well beyond the local existing networks. The initiative emerged from, and depended on, their enthusiasm and expertise. The network of trust of young person-orientated practitioners was then supported by key actors within wider management. This led to an on-going training programme that sought to make the young people's healthcare visible across the organisation, to begin to get to people to reappraise values and commitment.

## DISCUSSION

Across each organisation a complex, distributed, network of adolescent-oriented practitioners understood the potential value and worth of practices and services for young people, such as DAH. Within and across organisations, there is an informal knowledge economy of young people-oriented practitioners, teams and spaces. People have a preference for referring to other young person-orientated practitioners or spaces, to others within their networks of trust. Such referrals help to create, sustain and reinforce the network over time. As such, we see evidence of strong communities of practice,[29] focused on supporting and enacting adolescent medicine. table 2 illustrates our results in relation to the four theoretical constructs of NPT.[21]

The potential for normalisation of DAH is high within the group of young person-oriented practitioners and managers, as they make sense of, buy-into, enact, and evaluate it as worthwhile. Beyond the networks of young person-oriented practitioners and managers, we observed a low potential for normalisation. There are differences in opinion about the meaning and worth of DAH, its organisational and policy relevance, its potential workability, and its resource and spatial allocation. At one hospital, providing an organisation-wide strategy and training offered a clear opportunity to increase the potential for normalisation of DAH. Although not mandated, this programme of training, with support to 'back fill' those attending it, offered the greatest potential for an increase in people seeing (aspects of) DAH as legitimate, to increase buy-in and enrolment. This training has the potential to further extend the networks of young person-oriented practitioners and managers. It relied on buy-in and formal support from senior managers in both child and adult services to initiate and sustain it. The other two sites lacked any formal policy initiatives and any formal support and so had, at the time of fieldwork, a low potential to transform values and commitment across their respective organisations. At the start of the fieldwork, there was no national guidance on DAH. However, since the fieldwork, national guidance on transition[18] states that such care should be developmentally appropriate. This has the potential to enable change. The formal and informal groups with an interest in young people's care at those sites would clearly need buy-in and formal support from senior managers.

**Table 2** The four constructs of normalisation process theory (NPT) mapped against practitioner groups

| NPT construct | Networks of young person-oriented practitioners and managers | Other practitioners and managers who work with young people |
|---|---|---|
| Coherence: do people make sense of DAH? | See how DAH extends and is related to other approaches to care of young people; relatively shared understanding of purpose of DAH; understanding of impact of DAH on their work and see potential value and worth of DAH | Diverse views on relationship to other approaches; lack of shared understanding of purpose of DAH; diverse understanding of potential impact of DAH on their work; uncertainty around of potential value and worth (especially, given competing demands) |
| Cognitive participation: do people get involved with providing DAH and stay committed? | They are the key people driving DAH forward; they see DAH as legitimate, generally core, part of role; are very willing to work with others to enable DAH and motivated to deliver DAH over time | Aware that key people are driving DAH forward (key young person-orientated practitioners); lack of agreement that DAH legitimate part of work; some are willing to work with others to enable DAH; some are motivated to deliver over time |
| Collective action: do people make DAH work in practice? | DAH is operationalisable, especially within network; trust people in network to enact DAH, but less trust beyond; right mix of skills and training to undertake DAH in network, again, less beyond; in one site, clear support for DAH in organisation | Diverse views on workability of DAH and on trust about whether the right people are enacting DAH; lack of skills to undertake DAH, with training offered a one site; in one site, clear support for DAH in organisation |
| Reflexive monitoring: do people evaluate DAH as worthwhile? | Aware of impact of DAH; assess DAH as worthwhile and individually assess DAH as working well; enact DAH flexibly | Unsure of impact of DAH; unsure of whether worthwhile (given competing demands) or working well in practice; may enact some elements of DAH flexibly |

DAH, developmentally appropriate healthcare.

This is the first ethnographic study addressing the area of DAH for young people. This study gathered considerable data from many sources. The focus on exploring the provision of healthcare for young people over time, within specific teams and specialities, as well as across an organisation, enabled us to demonstrate the diversity within and across professional, team, ward and organisational boundaries. However, we only focused on three different organisations. Notably, all these organisations had a reputation for undertaking research on the care young people, so may represent examples of 'good practice'. Given the timing of our fieldwork, we did not get the opportunity to observe the impact of the roll-out of training around DAH that occurred at the one of the NHS trusts. A potential limitation of the study is that no young people were interviewed. However, there is a large literature of young people's experience in hospital settings but much less literature focussing on the perspective of health professionals and particularly of hospital managers.

Policy and research has emphasised the centrality of service delivery and workforce capacity issues to achieve successful integration and consistency of health services for all young people across organisations.[8 9] The results presented here add to this evidence base as we observed how the community of practice, the networks of young person-orientated practitioners and managers also support and provide, through formal and informal means, training and development around young people's health. Previous research has shown that continuing medical education in adolescent health can increase developmentally appropriate practices (such as confidential services)

and has the potential to address systemic barriers to healthcare for young people.[30] Thus, the role of such networks may be vital to achieving any sustainable change in the provision of healthcare for young people. Across all the sites, young people's formal and informal champions were key to getting people involved in the healthcare of young people. Research has outlined the importance of staff attitudes as a key determinant of young people's satisfaction with care.[4 31] Prior research has also shown the problem of staff turnover, if services rely on key individuals.[32 33] As such, there needs to be consistency of approach to developing local cultures of good practice that can withstand changes in personnel. In primary care, the Adolescent Champion model, which trains a multidisciplinary team of practice staff to deliver training to other staff and implement local quality improvement, has shown potential for sustainable change.[34]

Our findings suggest that there is a need for strong and clear guidelines, strategies and policies on the practical implementation of DAH at three levels: those of the individual young person and their family, the multidisciplinary team level and the organisation and/or system level. Our research informed the development of a toolkit for implementation of DAH at these levels which includes description of the models of good practice in terms of multidisciplinary working and training at the three hospitals studied.[35 36] Since the fieldwork ended, additional guidance has been published including for the care of young people in acute care settings[37] which includes a focus on DAH, alongside national guidance on the need to implement DAH in relation to transitional care.[18] A focus on specific settings (like acute

care) or contexts (like transition) has limitations as it can introduce or sustain inequities. Buy-in and formal support from senior managers in both child and adult services seems essential. Providing an institution-wide strategy and training seems key, particularly in view of the current unmet training needs reported in both paediatric[38] and adult physicians.[39] Currently adolescent and young adult medicine is not a recognised discipline in its own right in the UK unlike Australia and North America[40] although even when it is an established discipline, challenges remain, for example in the USA.[41] The 'informal' adolescent and young adult medicine community in the UK may also need to become more formally recognised. Young people should no longer be seen as 'vary rarely ill' or allowed to remain 'just below the radar', or seen as 'too old' for child services or 'too young' for adult. Core principles underlying the practice of adolescent medicine, such as DAH, should not remain contested. We need to enable people to reappraise values and commitment, to understand them as a normal part of everyday service provision.

**Author affiliations**
[1]Department of Social Work, Education and Community Wellbeing, Northumbria University, Newcastle upon Tyne, UK
[2]Institute of Health and Society, Newcastle University, Newcastle upon Tyne, UK
[3]School of Nursing and Health Sciences, University of Dundee, Dundee, UK
[4]Institute of Neuroscience, Newcastle University, Newcastle upon Tyne, UK
[5]Department of Applied Health Research, University College London, London, UK
[6]Child Health Department, Northumbria Healthcare NHS Foundation Trust, North Shields, UK
[7]Arthritis Research UK Centre for Epidemiology, Centre for Musculoskeletal Research, University of Manchester, Manchester, UK
[8]NIHR Manchester Biomedical Research Centre, Manchester University NHS Foundation Trust, Manchester, UK

**Acknowledgements** We acknowledge the support of Dr Elizabeth Rankin for her support and the health professionals and staff at the three participating hospitals. We also thank all the members of the youth advisory group—UP—for their support and advice.

**Collaborators** Tim Rapley, Albert Farre, Jeremy R Parr, Victoria J Wood, Debbie Reape, Gail Dovey-Pearce, Janet McDonagh, Allan Colver, Helen McConachie, Mark S Pearce, Luke Vale, Caroline Bennett, Greg Maniatopoulos; advisors: Nichola Chater, Helena Gleeson.

**Author Contributions** TR, JMcD, JRP and DR designed the original study. VJW and AF carried out data collection and data analysis under the supervision of TR, JMcD, DR, GD-P and JRP. TR and AF led on the writing of this manuscript. All authors worked on drafts of the paper and approved the final version of this article.

**Funding** This article presents independent research funded by the National Institute for Health Research (NIHR) under the Programme Grants for Applied Research programme: RP-PG-0610-10112. The views expressed in this article are those of the authors and not necessarily those of the NHS, the NIHR or the Department of Health. The funder took no part in the collection, analysis or interpretation of the data, in the writing of the article nor in the decision to submit the article for publication.

**Competing interests** None declared.

**Patient consent for publication** Not required.

**Ethics approval** The study received a favourable opinion from the National Research Ethics Committee (12/NE/0423).

**Provenance and peer review** Not commissioned; externally peer reviewed.

**Data availability statement** No additional data are available. Data are difficult to anonymise and we had not sought permission for data sharing.

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
