## [Reviewer comments · BMJ Open]

ARTICLE DETAILS

TITLE (PROVISIONAL)	Can we normalize developmentally appropriate health care for young people in UK hospital settings? An ethnographic study
AUTHORS	Rapley, Tim; Farre, Albert; Parr, Jeremy; Wood, Victoria; Reape, Debbie; Dovey-Pearce, Gail; McDonagh, Janet

VERSION 1 - REVIEW

REVIEWER	Natàlia Carceller-Maicas Medical Anthropology Research Center. Department of Anthropology, Philosophy and Social Work. Universitat Rovira i Virgili. Tarragona. Spain.
REVIEW RETURNED	16-Feb-2019

GENERAL COMMENTS	COMMENTS FOR THE AUTHORS The topic that approaches this paper is interesting and innovative. Medical organizations and practitioners are not used to put the focus on young people, and usually they have an adult-centric perspective. One approach like this that takes in care the young people point of view, their experiences, and their needs and difficulties during the process of health-illness-attention is needed in order to provide the best attention to this age group. Approach this topic using a qualitative methodology doing observation and interviews offers the possibility to obtain accurate information about the opinions, feelings, subjectivities and problems that institutions and professionals find when they want to approach youth illness from a new non-adult-centric perspective. Explore 3 places that work in a young people oriented way offers the opportunity to analyze in deep the processes they have followed, the problems they have found and the ways in which they have solved it, which offers the possibility of sum up in this paper some recommendations of best practices that could be used by other institutions interested in implement this practice. In order to publish this manuscript my recommendation is that major revisions are needed. Recommendations are listed below, and are organized by parts according to the paper's order and structure. Include these recommendations would improve the final writing, clarifying the research process, making replicable this research, and adding a final practical oriented point with best practice recommendations. ABSTRACT: Participants: 192 participants including health professionals (n=103) and managers (n= 72) were recruited.
--

What about the other people? $103+72= 175$. What about the other participants? You need to include them and their role at this point of the abstract.

It could be interesting include which are the specialties of these professionals. (Diabetes, Emergency Care, General Paediatrics, Outpatients, Rheumatology, and Trauma and Orthopaedics).

There is no information about fieldwork and about which techniques you have used to obtain information. You must include also when you start and finish the research.

It could be interesting to include more information about location, because you only say "England", and not all places and regions have the same socio-demographic reality and particularities, and that must be taken into account.

METHODS:

It is needed that you explain why if you have 192 participants you had only 65 formal qualitative interviews. Why did you interview only 65 persons? Why not the entire sample?

You need to explain why you used different techniques to interview people. Why some people by email, other by phone, and other in person?

How much people did you interview in each way?

Have you found differences in the results and information obtained depending on the way you make the interview?

How many participants did you interview in each interview? Each one of those 65 interviews was addressed to one person, or maybe to 2 or more participants?

Have you used different guides of interview to different specialties or have you use the same model for all professionals?

What was the participation of the other 127 persons? You must explain this properly.

About the 1600 hours of non-participant observations you need to explain where was it conducted. Did you make the observation at the hospital rooms? Did you make it at medical consultation room, or maybe at the hospital hall?

Why only 48 participants were both observed and interviewed?

What about the other persons interviewed? Why they weren't observed?

There isn't information about the settings and locations. You only say "England", and not all places and regions have the same socio-demographic reality and particularities, and that must be taken into account. Are the hospitals in a rural area or in a capital city? And about the installations, are old or is a new hospital with young and new professionals? Have you analyzed these points? Is there any relevant difference between the 3 institutions in this term?

RESULTS AND DISCUSSION

Why young people were not interviewed?

If you only interview professionals you are losing the patients' point of view and their experience with the institution, with the professionals, and about their entire process inside the hospital, which is especially interesting in a qualitative study like this.

A reflection and an adequate discussion about this point as a limitation of the study are needed.

How many years were those professionals and managers working in this way? Which kind of specific formation in young people have they (in case they have it)? Have you found gender and/or age differences among participants?

	Have you found differences between the professionals of several specialties? Which specialties are more young people enthusiastic or oriented? Which specialties is less? You speak about groups of interest, enthusiastic of young people. It could be really interesting to explain more about how all this groups start in each one the three spaces, talk about the reasons that motivate the creation of these groups, the several reactions that the organization/hospital have about this new groups (if it was well received, if they support it or encourage it, etc.), the way they organize their meetings and how they work in their multidisciplinary teams, explain more about the role that mental health and emotions professionals have in these groups, explain more about how these groups deal with families and with patients. It could be interesting to include a point in the discussion with some recommendation for best practices addressed for other medical organizations, in order to encourage and facilitate the implementation of this approach.
--	---

REVIEWER	Graham Martin University of Cambridge
REVIEW RETURNED	22-Feb-2019

GENERAL COMMENTS	This appears to be a well conducted study of the realisation of developmentally friendly services for young people aged 10-24 in three hospitals with varied characteristics in England. While its findings and recommendations are not especially surprising (e.g. the need for senior buy-in, the value of an informal community of practice that focuses on high-quality care for young people, and the presence of informal networks of referral and knowledge exchange that link into and out of that community), it does offer an interesting case study that will be of value to people interested in this field, and in the challenges of spreading and embedding particular forms of care seen a desirable but lacking in prioritisation. I have three major suggestions and several minor, largely presentational ones. First, I was not clear on what the impetus was (if any) towards developmentally appropriate healthcare in the three hospitals examined. Were they actually trying to implement this form of service delivery? Was it something that was mandated (or at least valued) top-down? Were particular groups keen to put it into practice? Was it associated with any budgets or targets or specific pathways? This seems particularly important in understanding and interpreting the data and analysis on legitimacy and buy-in / enrolment. To what extent was developmentally appropriate healthcare something that all three hospitals were seeking to promote, and what forms did this promotion take? Or indeed if there was an absence of such a push – i.e. DAH was a high-level (e.g. WHO) discourse that had no national policy or local organisational support, and was left purely to the advocates and enthusiasts to push forward – then this too is an interesting feature of the case studies, and is worth explaining explicitly, and examining in terms of its influence on the process of patchy normalisation that the authors find. Some of this becomes clearer very late on in the course of the presentation of the findings, but I think it would be helpful to foreground it and hang some of the analysis off this issue, including the ways in which the three hospitals differed, and the consequences of this for the realisation of developmentally appropriate healthcare.
---

	Second, I felt in the introduction that the authors could perhaps provide a bit more detail on what distinguishes a 'youth friendly' system from a 'developmentally appropriate' / 'adolescent-responsive' healthcare system – i.e. what do 'youth-friendly' systems lack that adolescent-responsive systems possess? Third, I found it slightly strange that despite the extensive ethnographic observation mentioned in the methods, the findings section seems to rely exclusively on quotations from interviews for data excerpts. I know that there are sometimes good reasons for this—interview quotations can much more pithily sum up a key theme—but it probably deserves acknowledgement and justification. Minor comments There are some inconsistencies of tense across the paper, e.g. veering into present tense page 10 lines 12-54 among other points, which make the paper a little difficult to read Page 11 lines 30-32: "Exposure to working with young people was central to enabling young people to become seen as just another 'young patient', over a set of unknown and unexpected concerns." I did not understand this sentence. Page 12 line 30: were should read was Page 12 line 37: where should read were Page 13 line 24: every day should read everyday
--	--

VERSION 1 – AUTHOR RESPONSE

Reviewer: 1

ABSTRACT:

- Participants: 192 participants including health professionals (n=103) and managers (n= 72) were recruited. What about the other people? $103+72= 175$. What about the other participants? You need to include them and their role at this point of the abstract.

We apologise this was our mistake. We went back over our data logs to re-check the numbers. We recruited n=121 health professionals and n=71 managers, so a total of n=192 people recruited. Of the health professionals, 85 were recruited for observation only, 23 for interview only and 13 for interview and observation (and of these 13, 8 were formally interviewed twice). Of the managers, 55 were recruited for observation only, 7 for interview only and 9 for interview and observation (and of these 9, 5 were formally interviewed twice). This has now been rewritten as follows:

'Participants 192 participants, health professionals (n=121) and managers (n= 71) were recruited between June 2013-January 2015. Approximately 1600 hours of non-participant observations in clinics, wards and meeting rooms were conducted, alongside 65 formal qualitative interviews.'

- It could be interesting include which are the specialties of these professionals. (Diabetes, Emergency Care, General Paediatrics, Outpatients, Rheumatology, and Trauma and Orthopaedics).

As the abstract was limited to 300 word count these details were not included here.

- There is no information about fieldwork and about which techniques you have used to obtain information.

Given the 300 word limit for the abstract, we only have space to include the information that the fieldwork comprised of non-participant observations, with some examples of the spaces we worked in, and interviews. This has now been rewritten as follows:

'Participants 192 participants, health professionals (n=121) and managers (n= 71) were recruited between June 2013-January 2015. Approximately 1600 hours of non-participant observations in clinics, wards and meeting rooms were conducted, alongside 65 formal qualitative interviews.'

- You must include also when you start and finish the research.

We have now included the start and finish date.

'Participants 192 participants, health professionals (n=121) and managers (n= 71) were recruited between June 2013-January 2015. Approximately 1600 hours of non-participant observations in clinics, wards and meeting rooms were conducted, alongside 65 formal qualitative interviews.'

- It could be interesting to include more information about location, because you only say "England", and not all places and regions have the same sociodemographic reality and particularities, and that must be taken into account.

Further detail was not added to the abstract due to the limitations of the word count.

METHODS:

- It is needed that you explain why if you have 192 participants you had only 65 formal qualitative interviews. Why did you interview only 65 persons? Why not the entire sample?

Those that we choose to interview were central to us developing our understanding of the work and furthering our conceptual ideas. When undertaking such large scale ethnographic work, we would argue that it is very rare that you would plan to conduct formal, audio-recorded, interviews with every participant that you observe. However, in the process of periods of observation, at times, we did talk

with those that we have observed. Those informal conversations were not audio-recorded, but rather were used to further inform our fieldnotes

- You need to explain why you used different techniques to interview people. Why some people by email, other by phone, and other in person? How much people did you interview in each way?

We are sorry for the confusion. All participants were interviewed the same way, face-to-face. The various strategies mentioned in the text refer to recruitment strategies, not the practices of interviewing. Participants were approached to recruit them using a variety of methods including face-to-face, telephone and email. The following sections of the Methods has been revised:

'Participants were approached to take part in the study using a variety of methods including face-to-face, telephone and email.

- How many participants did you interview in each interview? Each one of those 65 interviews was addressed to one person, or maybe to 2 or more participants?

Formal interviews were all one-to-one and all conducted face-to-face. The text has been revised as follows:

'Formal interviews (average length, 45 minutes) were conducted face-to-face on a one-to-one basis and were audio-recorded, transcribed, edited to ensure respondents anonymity and then analysed alongside anonymised fieldnotes.'

- Have you found differences in the results and information obtained depending on the way you make the interview?

As noted above, all the interviews were conducted the same way.

- Have you used different guides of interview to different specialties or have you use the same model for all professionals? What was the participation of the other 127 persons? You must explain this properly.

Participants were either:

(1) observed undertaking their normal work practices (n=140)

(2) formally interviewed about their work practices (n=30)

(3) both observed undertaking their work practices and formally interviewed about their work (n=22; note that of these 22, n=13 took part in two interviews)

We have amended Table 2, to make this clear.

Initial topic guides were designed for two groups 'health professionals' and 'managers'. The guides were designed to be used across specialities and areas. The guides evolved as result of analysis, allowing for tailoring and gradual integration of a range of issues and topics of relevance to specific roles, settings, specialities or areas that we wanted to explore further.

The following has been added to the text:

'Initial topic guides were designed for clinicians and managers irrespective of setting and evolved during the course of data collection, allowing for tailoring and gradual integration of a variety of follow-up issues and topics of relevance to specific roles, settings, specialities or areas.'

- About the 1600 hours of non-participant observations you need to explain were was it conducted. Did you make the observation at the hospital rooms? Did you make it at medical consultation room, or maybe at the hospital hall?

The observations were conducted within and across a wide variety of hospital spaces, depending on the nature of the session and/or the professionals involved (e.g. consultation rooms, wards, meeting rooms, public spaces, movement between separate organisational settings). The following text has been added:

'Observations were conducted within and across a wide variety of hospital spaces – including clinics, wards and meeting rooms - depending on the nature of the session and/or the professionals involved. They were recorded in contemporaneous fieldnotes'.

- Why only 48 participants were both observed and interviewed? What about the other persons interviewed? Why they weren't observed?

As with the interviews, observations were only conducted involving specific individuals as and when relevant, based on the type of data that we sought to collect from a given participant and/or setting and the data previously collected and analysed. For example, some interviewees worked in areas where we were already observing through another professional and where further observations from an already well documented perspective were not considered necessary.

- There isn't information about the settings and locations. You only say "England", and not all places and regions have the same socio-demographic reality and particularities, and that must be taken into account. Are the hospitals in a rural area or in a capital city? And about the installations, are old or is a new hospital with young and new professionals? Have you analyzed these points? Is there any relevant difference between the 3 institutions in this term?

Due to the nature of the study and to ensure honest responses from participants, the hospitals were not named but they were all large, well-established hospitals in urban settings outside London. One was a stand-alone children's hospital providing tertiary paediatric care, one a large adult hospital without paediatrics providing tertiary adult care and one was a secondary care hospital providing both paediatric and adult care. The core analytic issues was to focus on the work with young people. This did not differ in relation to geographical factors (but, we note that, we only focused on two regions). In term of age of building and staff, all had professionals from a wide age range and a range of ages of buildings. In the UK, it is relatively rare for a new hospital to be built. They may add new buildings on the existing site, or sometimes on an additional site, but they are always served by a mixture of staff in terms of experience and age. The following text has been added:

'This ethnographic study was conducted across three hospitals in two regions of England (a district general hospital, a large paediatric tertiary hospital and a large adult tertiary hospital) all in urban settings outside London'.

RESULTS AND DISCUSSION

- Why young people were not interviewed? If you only interview professionals you are losing the patients' point of view and their experience with the institution, with the professionals, and about their entire process inside the hospital, which is especially interesting in a qualitative study like this. A reflection and an adequate discussion about this point as a limitation of the study are needed.

We wholeheartedly agree with the reviewer that the perspective of young people should always be considered when considering health care provision. There is a large literature of young people's experience in hospital settings (and some authors of this paper have been involved in such work). However, we were very aware that there is much less literature focussing on the perspective of health professionals, and particularly of hospital managers and looking across an organisation, hence the design of this study. The following text has been added:

'A potential limitation of the study is that no young people were interviewed. However there is a large literature of young people's experience in hospital settings but much less literature focussing on the perspective of health professionals and particularly of hospital managers.'

- How many years were those professionals and managers working in this way? Which kind of specific formation in young people have they (in case they have it)? Have you found gender and/or age differences among participants? Have you found differences between the professionals of several specialties? Which specialties are more young people enthusiastic or oriented? Which specialties is less?

This question is a really interesting research question. Within our data set, there was no distinct pattern to being more young people enthusiastic or oriented that we would want to align with training, gender, age, timing or speciality. Clearly, different areas/specialties will have different resources, focus and histories of working with young people (given the conditions they focus on), but nothing clear emerged. So, we routinely saw that, for example, that within one paediatric or adult speciality at one hospital, staff had different ways of working with young people. Also, the same specialty across all three organisations had a different focus on young people. When specific staff leave, that focus can be lost. As we note in the paper, some specific wards, notably the charity-supported oncology wards, are designed and resourced solely for young people. So, no specific pattern emerged in the data to say that a particular type of person or speciality that is more likely. To generate a data-set to focus on this question, we feel we could work with one or more of the adolescent and/or young people special interests groups that many of the professional societies have, in order to explore, across a large data set, those factors that may align people with this interest.

- You speak about groups of interest, enthusiastic of young people. It could be really interesting to explain more about how all this groups start in each one the three spaces, talk about the reasons that motivate the creation of these groups, the several reactions that the organization/hospital have about this new groups (if it was well received, if they support it or encourage it, etc.), the way they organize their meetings and how they work in their multidisciplinary teams, explain more about the role that mental health and emotions professionals have in these groups, explain more about how these groups deal with families and with patients.

We agree, understanding in detail about how these groups form and develop, is a really interesting topic. We are actually planning on doing some further work and a paper on this topic, as this has the potential to support further change. As we note in the paper, they have different levels of buy-in from senior management, different levels of reach. Several of these groups are described in a key output of this research in the form of a toolkit for health professionals and managers [35]. We are limited by what we can focus on by the word count, so refer people to the toolkit to gain access to some of the information you outline. The following has now been added to the revised draft to highlight the availability of such detail:

'Our research informed the development of a toolkit for implementation of DAH that includes description of the models of good practice in terms of multidisciplinary working and training [35,36].'

- It could be interesting to include a point in the discussion with some recommendation for best practices addressed for other medical organizations, in order to encourage and facilitate the implementation of this approach.

As we noted in response to your last comment, we have sought to do this by referring to Due to lack the toolkit which was informed by this research and includes examples of such models of good practice [32,35,36].

Reviewer: 2

- I was not clear on what the impetus was (if any) towards developmentally appropriate healthcare in the three hospitals examined. Were they actually trying to implement this form of service delivery? Was it something that was mandated (or at least valued) top-down? Were particular groups keen to put it into practice? Was it associated with any budgets or targets or specific pathways? This seems particularly important in understanding and interpreting the data and analysis on legitimacy and buy-in / enrolment. To what extent was developmentally appropriate healthcare something that all three hospitals were seeking to promote, and what forms did this promotion take? Or indeed if there was an absence of such a push – i.e. DAH was a high-level (e.g. WHO) discourse that had no national policy or local organisational support, and was left purely to the advocates and enthusiasts to push forward – then this too is an interesting feature of the case studies, and is worth explaining explicitly, and examining in terms of its influence on the process of patchy normalisation that the authors find. Some of this becomes clearer very late on in the course of the presentation of the findings, but I think it would be helpful to foreground it and hang some of the analysis off this issue, including the ways in which the three hospitals differed, and the consequences of this for the realisation of developmentally appropriate healthcare.

All three sites had a history of championing research and innovative service provision for young people. However, at the start of the fieldwork one site – the General Hospital - was developing a policy about DAH. This had emerged from a formal, organisationally-supported, group that focused on the care of young people, that was initially formed to focus on transition. This strategy group comprised of managers and senior clinicians with a strong interest in young peoples health. At another site - the Paediatric Hospital - there was no explicit policy on DAH, but one focused on transition. They also had a formal, organisationally-supported, group focused on young people. However, thinking about DAH was being driven by advocates and enthusiasts alone. Finally, the third site had no policy on DAH, transition or young people. The sole trust-wide initiative is an informal interest group focused on young people issues.

We have three different contexts in which this work is going on, with different layers of formal and informal organisational support, influence and reach. For those outside the networks of young person-oriented practitioners and managers, we observed a low potential for normalization. Unsurprisingly, the site with the formal policy, which had begun to deliver the training, which had supported time to be released for training, offers the greatest potential for an increase in people seeing (aspects of) DAH as legitimate, to increase buy-in and enrolment. The other two sites, those without a formal policy, formal support, and in one site, only an informal interest group, have a low potential to transform values and commitment. At the start of the fieldwork, there was no national guidance on DAH. However, since the fieldwork, DAH is now part of national guidance on Transition. This has the potential to enable change. The formal and informal interest groups could refer to these new standards to leverage some interest, to enable buy-in and formal support senior managers.

The following has been added to the methods section:

‘All hospitals had a history of championing research and innovative service provision for young people. At the start of the fieldwork one site – the General Hospital - was developing a policy about DAH. This had emerged from a formal, organisationally-supported, young people’s group, which was initially formed to focus on transition. This strategy group comprised of managers and senior clinicians. At another site - the Paediatric Hospital - there was no explicit policy on DAH, but one focused on transition. They also had a formal, organisationally-supported, young peoples group. However, thinking about DAH was being driven by advocates and enthusiasts alone. Finally, the third site had no policy on DAH, transition or young people. The sole organisation-wide initiative is an informal interest group focused on young people issues.’

The following has been added to the discussion section:

‘At one hospital, providing an organisation-wide strategy and training offered a clear opportunity to increase the potential for normalization of DAH. Although not mandated, this programme of training, with support to ‘back fill’ those attending it, offered the greatest potential for an increase in people seeing (aspects of) DAH as legitimate, to increase buy-in and enrolment. This training has the potential to further extend the networks of young person-oriented practitioners and managers. It relied on buy-in and formal support from senior managers in both child and adult services to initiate and sustain it. The other two sites lacked any formal policy initiatives and any formal support and so had, at the time of fieldwork, a low potential to transform values and commitment across their respective organisations. At the start of the fieldwork, there was no national guidance on DAH. However, since the fieldwork, national guidance on transition [18] states that such care should be developmentally appropriate. This has the potential to support young person-oriented practitioners arguments for change. The formal and informal groups with an interest in young peoples care would clearly need buy-in and formal support from senior managers to introduce and embed wider change.’

- Second, I felt in the introduction that the authors could perhaps provide a bit more detail on what distinguishes a ‘youth friendly’ system from a ‘developmentally appropriate’ / ‘adolescent-

responsive' healthcare system – i.e. what do 'youth-friendly' systems lack that adolescent-responsive systems possess?

Youth-friendly healthcare has typically been enacted in terms of specific, localised space or team of people – for example, a single oncology ward, or, a single GP practice. The move towards adolescent-responsive healthcare systems is aligned with a focus that young people, as they move through life, work with a range of health care providers – for example, in one visit to a hospital, they may go to a specific clinic, then on a ward, then see physiotherapy, visit the pharmacy etc. In this way, we need to move beyond localised pockets of good practice, situated in specific spaces or within specific teams. Instead, we need to enable such good practice to be embedded across the system, to be embedded not only in specific spaces, or clinical encounters, but across organisations, embedded in all organisational strategies and policies.

The following has been added to the text:

'However, as highlighted by the WHO, there is the need to move from the ad hoc provision of youth-friendly healthcare services – often embedded in specific locations or teams - to adolescent-responsive healthcare systems [8]. All aspects of health and social care that young people engage with, the range of providers, organisations and policies, need to more responsive to and aligned with the care of young people'.

- Third, I found it slightly strange that despite the extensive ethnographic observation mentioned in the methods, the findings section seems to rely exclusively on quotations from interviews for data excerpts. I know that there are sometimes good reasons for this—interview quotations can much more pithily sum up a key theme—but it probably deserves acknowledgement and justification.

As the reviewer suggests, the interview quotes offered a shorthand way to express the key issues. The fieldnotes were central to shaping the analytic ideas, but take more words to demonstrate the key issues at hand.

The following text has been added:

'In presenting the analysis, we have drawn on interview quotes, over excerpts from fieldnotes, as they offer the reader rapid access to the key analytic ideas'.

Minor comments

- There are some inconsistencies of tense across the paper, e.g. veering into present tense page 10 lines 12-54 among other points, which make the paper a little difficult to read

- Page 11 lines 30-32: "Exposure to working with young people was central to enabling young people to become seen as just another 'young patient', over a set of unknown and unexpected concerns." I did not understand this sentence.
- Page 12 line 30: were should read was
- Page 12 line 37: where should read were
- Page 13 line 24: every day should read everyday

The paper has been revised as suggested

VERSION 2 – REVIEW

REVIEWER	Graham Martin University of Cambridge, UK
REVIEW RETURNED	03-Apr-2019

GENERAL COMMENTS	The authors have responded appropriately to the reviewers' comments, and the manuscript has improved correspondingly. I picked up one typo: "peoples" should read "people's" at the bottom of page 16.
--

VERSION 2 – AUTHOR RESPONSE

Reviewer 2

- The authors have responded appropriately to the reviewers' comments, and the manuscript has improved correspondingly. I picked up one typo: "peoples" should read "people's" at the bottom of page 16.

The paper has been revised as suggested